

# Operations and Maintenance Cost Comparison Between 15MW Direct-Drive and Medium-Speed Offshore Wind Turbines

Orla Donnelly, Fraser Anderson, and James Carroll

University of Strathclyde, 99 George Street, Glasgow G1 1RD

**Correspondence:** Orla Donnelly (orla.donnelly@strath.ac.uk)

**Abstract.** Determining offshore wind energy operational expenditures relies on acquiring reliability data, particularly as turbine power ratings increase. The uncertainty surrounding operational costs and optimal drive train configurations for larger turbines persists. While previous research addressed reliability data for 3 MW offshore wind turbines, this study reviews and collates updated failure data for 15 MW turbines, comparing direct drive and medium-speed configurations. It employs an Operations and Maintenance (O&M) modeling tool to calculate total operational costs. The study concurs with existing literature, showing that direct drive turbines have lower operational costs than medium-speed turbines in three case studies. However, the cost and availability differences between configurations are smaller than previously suggested. For 15 MW turbines, the analysis reveals that the cost disparity between direct drive and medium-speed turbines is significantly smaller than for smaller-rated turbines, with percentages of 1.59 %, 1.58 %, and 5.78 % for the three ScotWind sites selected. Previously, the absolute % difference in cost between direct drive and medium speed turbines was estimated at 29.79 %. Sensitivity analyses explore the influence of three factors—failure rates, accessibility limits, and major replacement times—on total operational costs. These analyses demonstrate that medium-speed configurations exhibit more significant cost fluctuations, and the cost gap between configurations is reduced if failure rates are lowered at the same rate for each configuration, accessibility increases or the major replacement time is reduced for the larger wind turbine components.

## 1 Introduction

With a strive to become net zero by 2050, the UK is continuing to invest more into the renewable energy industry (UK Government (2022)). Offshore wind is one of the predominant renewable energy resources that the UK has available, in particular in Scotland, with 10 year agreements in place to build 20 new offshore fixed and floating sites (Crown Estate Scotland (2023b)). To ensure the growth of the offshore wind market, further reduction in the cost of energy for wind energy is required in order to compete with other energy sources. Recent trends in offshore wind include projects that have larger wind farm sites, containing more wind turbines, which are increasing in rated nominal power (>5 MW) (Díaz and Guedes Soares (2020)). These sites are also now being located further from shore than pre-existing fixed bottom wind farms. These trends align with the Committee for Climate Change's (CCC) Net Zero report which calls for 75 GW to be installed offshore in the UK, which is around 7500 more turbines, rated at 10 MW, in order for net zero targets to be met (Committee for Climate Change (2021)). Due to the lack of available operational data for these sites, there remains large uncertainty around their operations and maintenance



(O&M) costs. In particular, there is a gap of knowledge surrounding generator and drive train technology in terms of reliability, diagnostics and prognostics. Previous research with old reliability data sets has provided insights into the operations and maintenance costs for smaller rated turbines with different drive train configurations, see Section 1.1. Collating and utilising newer failure data for the prominent drive train configurations used in larger rated turbines will provide an understanding of the

possible operations and maintenance costs for future wind farms, that are currently being developed. Modelling operations and maintenance for new wind farm sites allows comparisons to be made between drive train configurations in terms of availability, power production and total maintenance costs. Reducing unplanned downtime for turbines has potential to reduce the cost of generating offshore electricity by roughly 10 % (Carroll et al. (2017)).

This study provides novelty to the field, as it reviews literature for larger turbine failure rate estimates and data, putting some of it in the public domain through journal paper publication for the first time and uses O&M modelling to analyse:

- The impact of drive train type on operations and maintenance cost for 15 MW rated turbines.

- The sensitivity of O&M costs using lower and upper estimates of failure rates for 15 MW medium speed and direct drive turbines.

- The sensitivity of O&M costs by adjusting repair times for major components of a 15 MW rated turbine for different drive train configurations.

- The impact that accessibility for maintenance vessels has on the O&M costs for 15 MW turbines by altering the accessibility limits for the wind farm.

The rest of the paper is outlined as follows: Section 1.1 provides an overview of the medium speed geared turbine and direct

drive turbine, Section 2 provides a literature review of new reliability data for offshore wind turbines, Section 3 expands on the methodology used for the research, Section 4 contains the results of the baseline O&M scenario and several sensitivity analyses, Section 5 discusses the results, main assumptions and limitations of the work and Section 6 includes a conclusion and suggestion for further work.

## 1.1 Comparison of Geared and Direct Drive O&M Costs

Carroll et al. (2017), provide a comparison of O&M costs for offshore wind turbines with different drive-train configurations. To elaborate, the configurations are:

1. A 3-stage gearbox with a doubly-fed induction generator (DFIG) and partially-rated converter.

2. A 3-stage gearbox with a permanent magnet synchronous generator (PMSG) and fully-rated converter.

3. A 2-stage (medium-speed) gearbox with a PMSG and fully-rated converter.

4. A direct-drive turbine with a PMSG and fully-rated converter.





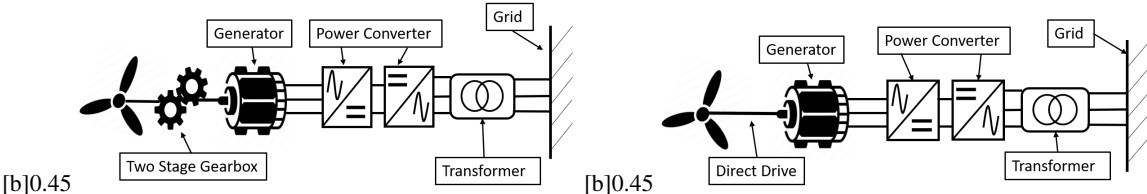

**Figure 1.** Drive train configurations for two offshore turbines. The left configuration shows a medium speed two stage permanent magnet generator with a fully rated converter. The right configuration is the direct drive turbine with a permanent magnet generator and full rated converter.

The results from this study show that the direct-drive configuration has the highest availability and lowest O&M costs, followed by the medium-speed configuration, then 3-stage with PMSG, then the DFIG. Lower O&M costs contribute to a lower overall Levelised Cost of Energy (LCoE) for direct-drive and medium-speed turbines (Carroll (2016)). This conclusion is corroborated by a trend in the industry to move away from older high-speed drive-trains and towards lower-speed machines. Figure 1 includes the two drive train configurations from Carroll (2016) that will be investigated in this study. Figure 2, taken

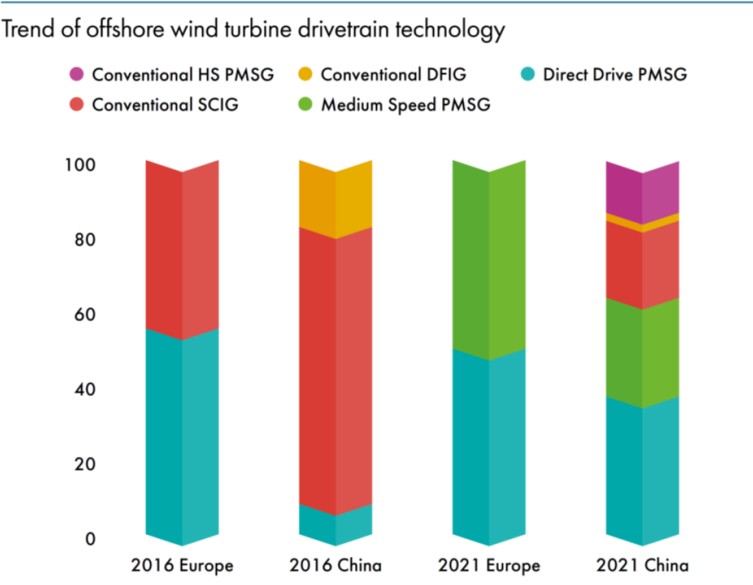

**Figure 2.** Offshore wind market share in Europe and China according to the GWEC's Global Offshore Wind Report 2022 (Global Wind Energy Council (2022)).


from the Global Wind Energy Council (GWEC)'s 2022 Global Offshore Wind Report shows the market share of the different drive-train concepts in 2016 and 2021 in both Europe and China. Europe has gone from an approximately equal split of conventional Squirrel Cage Induction Generators (SCIG) and Direct Drive PMSGs to an approximately equal split of Medium





Speed PMSG and Direct Drive PMSGs. China's proportion of low-and-medium-speed machines has also increased, however
around of the third of the market remains high-speed there.

From these figures, it seems the pertinent question in the industry has shifted from '*What is best out of high-speed/medium-speed/low-speed drive trains?*' to '*What is best out of medium-speed and low-speed drive trains?*'. A study by van de Kaa et al. (2020) presents a methodology to determine which of the two configurations has the highest chance of achieving success. They conclude that, as of 2020, both drive train types still have the potential to become dominant. They also conclude that the

energy cost and reliability are the most important determinants for success from a list also containing *brand reputation and credibility, total energy yield, pricing strategy, pre-emption of scarce assets, commitment* and *suppliers*.

Coinciding with this shift in the market towards direct drive turbines is a shift towards higher power ratings (Jenkins et al. (2022a)). This is a theme investigated by Turnbull et al. (2022). They pose the question: *how much insight can be drawn from data gathered on older technology and applied to modern direct-drive machines?* To answer this question, they present a

meta-analysis of the available literature. Their results show that, *"even for components that are assumed to be similar between turbine configurations, there are quite large differences in stop rate and downtimes"*.

Jenkins et al. (2022b), provide an estimate for availability and O&M costs from component replacements for the two configurations presented in Figure 1 for next generation (15 MW) turbines. Their replacement rates were based on structured expert elicitation, which is detailed further in Jenkins et al. (2022a). In contrast to the results of Carroll et al. (2017), their results

showed that medium-speed turbines had lower O&M costs for major replacements than direct-drive machines. They attributed the disparity to lower component costs of the medium-speed generator and gearbox compared to the expensive direct-drive generator.

## 2 Literature Review of New Reliability Data

The uncertainty of Carroll et al. (2017)'s estimates may be reduced by updated reliability data for offshore wind turbines in the

public domain. The data set presented by Carroll et al. (2016) and used in the original comparison of drive-train configurations remains the most comprehensive source of reliability figures for offshore wind turbines. Failure rates estimated in the study are summarised in Figure 3. The analysis also catalogues repair times, repair costs and number of technicians per repair for the same categories. However, Carroll et al. (2017)'s analysis has two disadvantages: (i) the failure data of Carroll et al. (2016) is becoming out of date with respect to the latest wind turbines and (ii) the availability and O&M rate estimates utilise some

transformed data from onshore turbines. Applying updated failure rates may reduce the uncertainty in the failure rate figures of the previous drive-train comparison. The following subsections elaborate on the sources for these updated failure rates.

### 2.0.1 Jenkins

Jenkins et al. (2022a) estimates replacement rates for major components of next generation turbines using the classical method of structured expert elicitation. Namely, they estimate replacement rates for the gearbox, generator and rotor for 15 MW

medium-speed and direct-drive turbines. Both fixed-bottom and floating turbine concepts are considered. While the cited





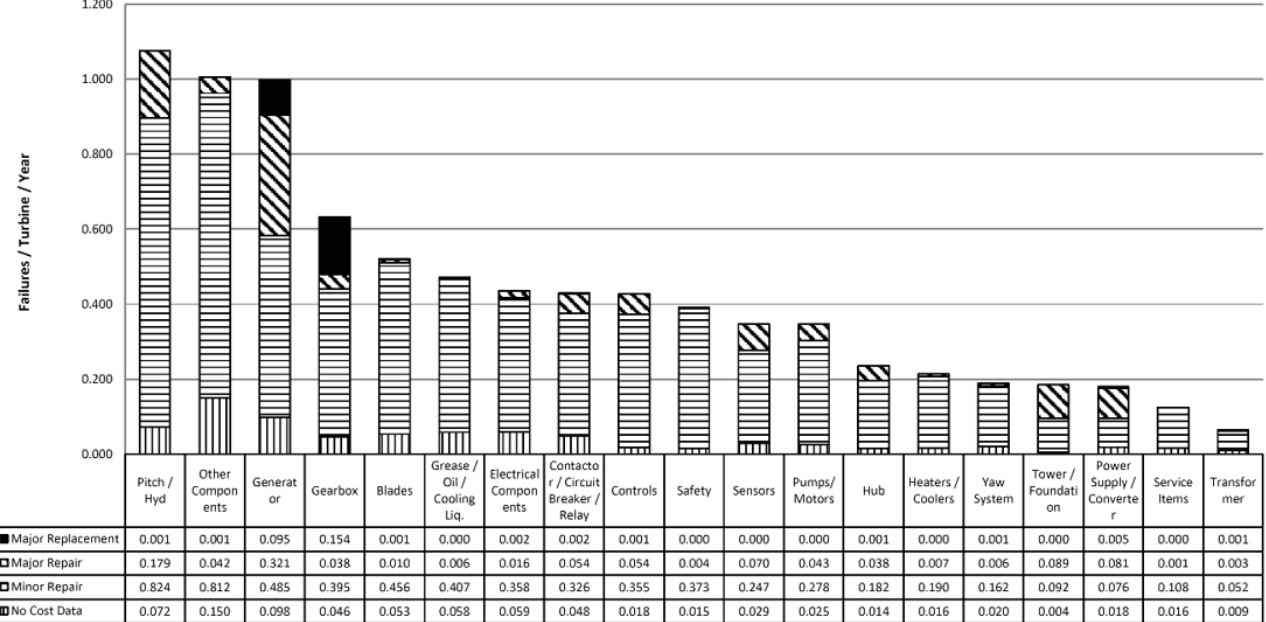

| | Pitch / Hyd | Other Components | Generator | Gearbox | Blades | Grease / Oil / Cooling Liq. | Electrical Components | Contactor / Circuit Breaker / Relay | Controls | Safety | Sensors | Pumps/ Motors | Hub | Heaters / Coolers | Yaw System | Tower / Foundation | Power Supply / Converter | Service Items | Transformer |
|---|---|---|---|---|---|---|---|---|---|---|---|---|---|---|---|---|---|---|---|
| ■ Major Replacement | 0.001 | 0.001 | 0.095 | 0.154 | 0.001 | 0.000 | 0.002 | 0.002 | 0.001 | 0.000 | 0.000 | 0.000 | 0.001 | 0.000 | 0.001 | 0.000 | 0.005 | 0.000 | 0.001 |
| ▨ Major Repair | 0.179 | 0.042 | 0.321 | 0.038 | 0.010 | 0.006 | 0.016 | 0.054 | 0.054 | 0.004 | 0.070 | 0.043 | 0.038 | 0.007 | 0.006 | 0.089 | 0.081 | 0.001 | 0.003 |
| ▤ Minor Repair | 0.824 | 0.812 | 0.485 | 0.395 | 0.456 | 0.407 | 0.358 | 0.326 | 0.355 | 0.373 | 0.247 | 0.278 | 0.182 | 0.190 | 0.162 | 0.092 | 0.076 | 0.108 | 0.052 |
| ▥ No Cost Data | 0.072 | 0.150 | 0.098 | 0.046 | 0.053 | 0.058 | 0.059 | 0.048 | 0.018 | 0.015 | 0.029 | 0.025 | 0.014 | 0.016 | 0.020 | 0.004 | 0.018 | 0.016 | 0.009 |

**Figure 3.** Failure rates for components by Carroll et al. (2016).

conference paper provides combined replacement rates for these components (Jenkins et al. (2022a)) , the breakdown of replacement rates by component is presented in Jenkins (2022) PhD thesis. The results of that analysis is shown in Figure 4.

Two sets of replacement estimates are provided by Jenkins (2022). These were obtained using different methods of structural expert elicitation, which is described in more detail in Jenkins (2022). The methodology has the benefit of uncertainty quantification, meaning that estimates for the $5^{th}$, and $95^{th}$ percentiles are available for each replacement rate estimate as well as the median. Key take-away points from Jenkins (2022) results are as follows:

1. Comparing the estimated failure rates for next-generation turbines shown in Figure 4 to first-generation turbines shown in Figure 3; there is a decrease in generator and gearbox major replacements and increase in rotor major replacements. The overall view of the experts used in that study is therefore that of a shifting risk profile. Major drive-train components have been identified as a problem point for first-generation turbines and consequently there has been an effort to increase their reliability.

2. Collectively, medium-speed drive-train components still have a higher replacement rates than direct-drive turbines.

### 2.0.2 Koukoura

Koukoura (2019) also includes drive-train failure rate data in their thesis. These are derived from field data. The field data is a population of 1200 offshore wind turbines from over 20 wind farms with a power rating between 2-10MW. It contains a





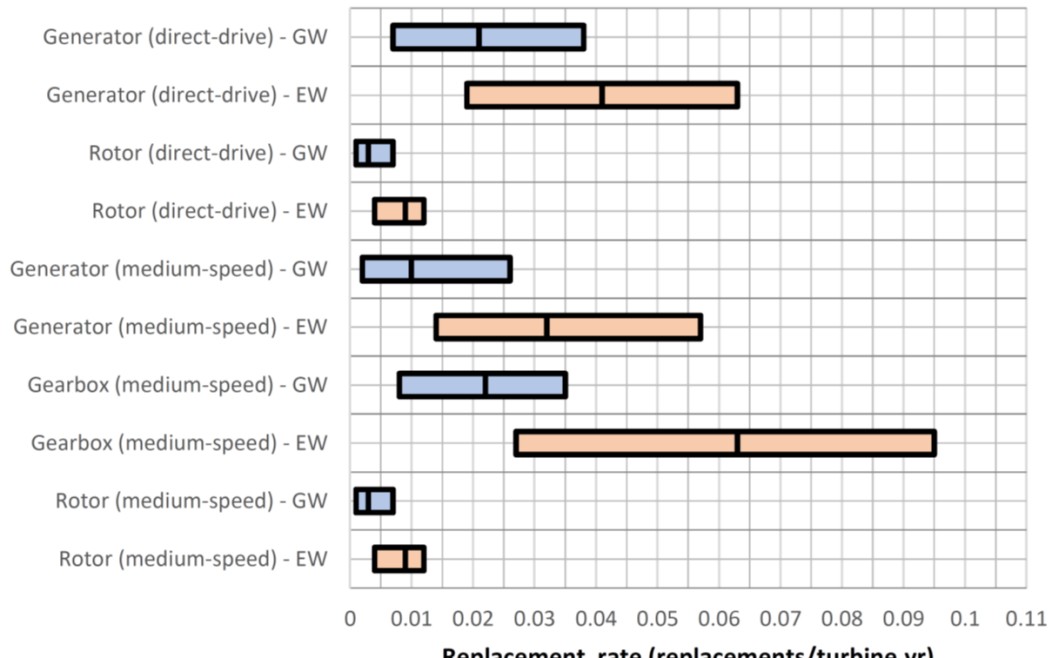

**Figure 4.** Major replacement rates for 15 MW fixed-foundation turbines, as presented by Jenkins (2022). *EW* stands for *Equal Weighting* and *GW* stands for *Global Weighting*. The $5^{th}$, $50^{th}$ and $95^{th}$ percentiles are shown for each component/elicitation method. Taken from Jenkins (2022).

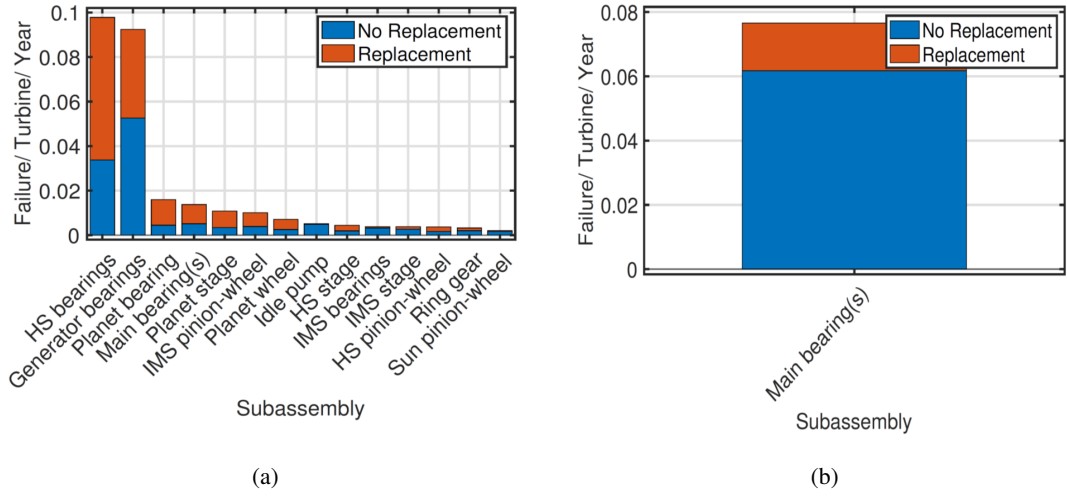

**Figure 5.** Mechanical drive-train failure rates for (a) geared turbines and (b) direct-drive turbines. Taken from Koukoura (2019).



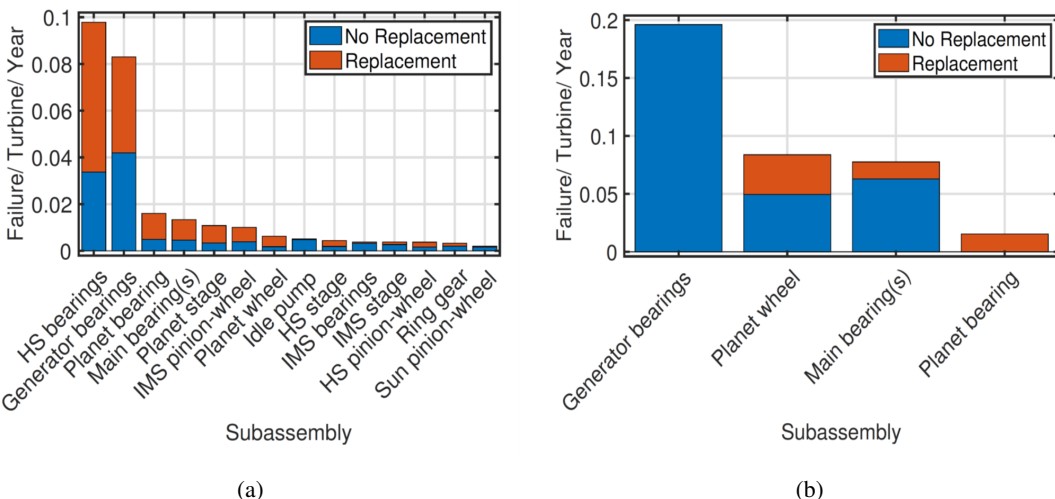

**Figure 6.** Mechanical drive-train failure rates for (a) turbines $< 5$ MW and (b) $\geq 5$ MW. Taken from Koukoura (2019).

mixture of drive-train configurations containing high-speed gearboxes, medium-speed gearboxes and direct-drive machines. They present two sets of partitioned data which might be of use in the following analysis. Namely;

1. they compare failure rates for direct-drive and geared turbines- Figure 5;

2. they compare lower power ratings ($< 5$ MW) to higher power ratings ($> 5MW$) - Figure 6.

The first point could ostensibly provide failure rate data which could be used directly for an O&M cost comparison. However, the population of geared turbines in that comparison contains a mix of high-speed and medium-speed gearboxes. The second point goes some way to breaking this down; newer, higher capacity turbines are more likely to be medium-speed than high-speed. Figure 6b also suggests a medium-speed population of turbines since many of the failure categories associated with a high-speed gearbox have been removed.

There are some features of Koukoura's failure rate figures that should be kept in mind if they are to be incorporated into the following analysis. First, the failures are mechanical only. For the generator, this effectively restricts the failure modes to the generator bearings. They do not capture failures in the generator fan, cooling system, stator/rotor issues or grease pipes. The gearbox and rotor are mechanical components, so most of the failure modes are captured for those components. Still, issues with the oil/lubrication system for those components are not captured. Second, the reliability statistics in Figures 5 and

6 are sub-assembly repairs/replacements. Only replacement rates for the gearbox are detailed in the thesis. These are shown in Figure 7.

Thirdly, both repair and replacement rates are presented for sub-assemblies. From a cost modelling perspective, this only provides one set of necessary inputs. Using these figures on their own would induce an uncertainty in the repair/replacement times and number of technicians needed for repair. Lastly, replacement and repair rates are presented for the main bearing.





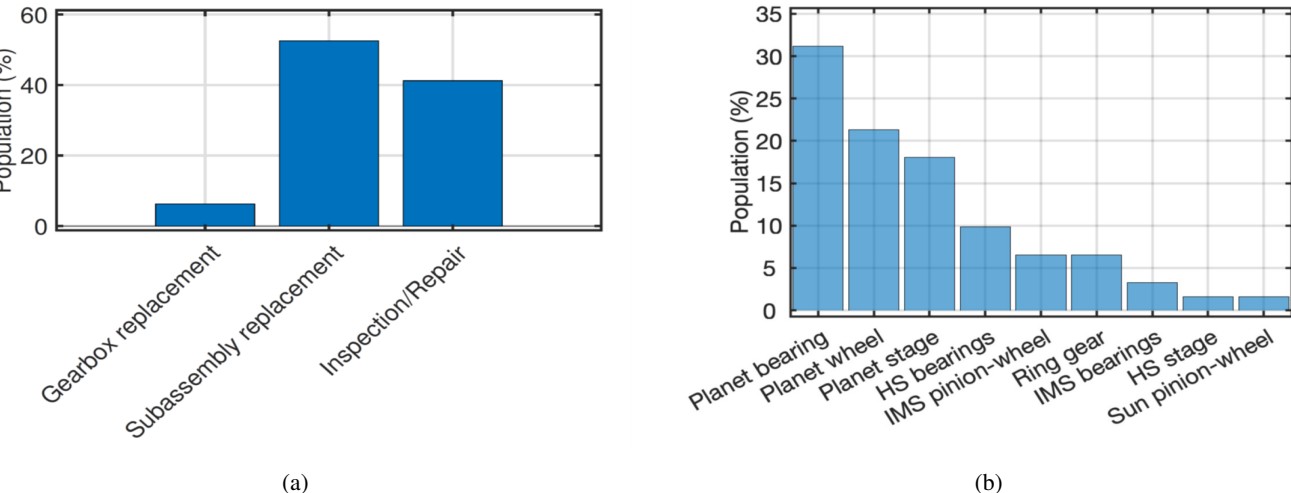

|                    (a)                    |                    (b)                    |

**Figure 7.** (a) Breakdown of gearbox replacements/sub-assembly replacements/repairs and inspections. (b) Breakdown of causes of gearbox replacement Koukoura (2019).

The main bearing is another key O&M cost driver in the drive train that was not considered as a separate component by either Carroll et al. (2017) or Jenkins et al. (2022a).

### 2.0.3 SPARTA

The SPARTA (System Performance, Availability and Reliability Trend Analysis) campaign is bench-marking initiative for offshore wind farms in the UK. The 20/21 SPARTA portfolio review by ORE Catapult (2022), contains major component

failure rates and forced outages per turbine. Their estimates for major component replacement rates are shown in Figure 8.

Again, these figures come with some caveats. Firstly, they represent a mixed population of first generation and current generation turbines which would contain high-speed, medium speed and direct-drive machines. It would also presumably contain a mix of predominantly first-generation DFIGs and current generation PMSGs. Secondly, SPARTA reports component repairs as "*forced outages*". How forced outages relate to repairs or failures in their definition is not clearly defined. As part of

that portfolio review they also compare the forced outages of direct drive turbines with geared turbines (grouped by capacity $< 3.6$ MW and $\geq 3.6$ MW). Figure 9 shows their results. Here we can see that turbines in the direct drive and $\geq 3.6$ MW categories show higher forced outage rates than in the $< 3.6$ MW category. However, SPARTA note that "*Since these turbines are still young, these failure rates can be expected to decrease in general in the future.*" and that "*more data is required to create strong insights about the differences between direct drive turbines and turbines with gearboxes.*"

### 2.0.4 Anderson

Anderson (2023), presents reliability data for an offshore wind farm in the UK. The analysed database consists of approximately 800 turbine hours of data. Reliability information is presented in terms of *interventions per year* in the case of non-corrective



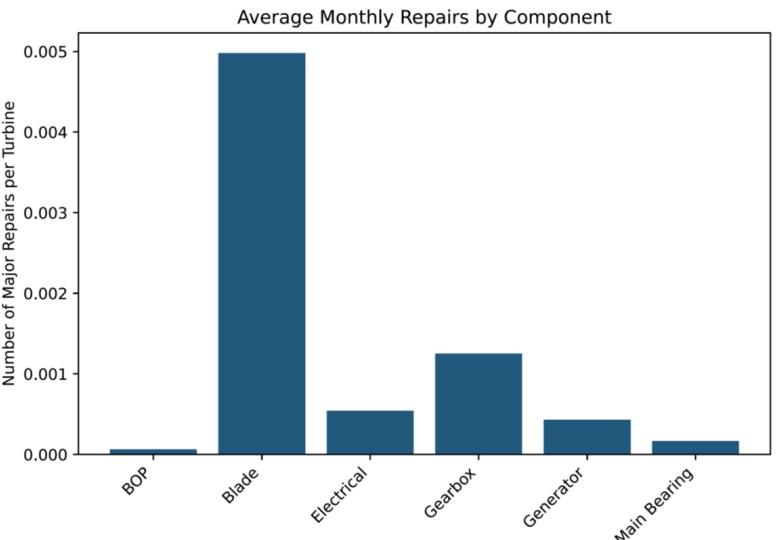

**Figure 8.** Major component replacement rates from the 20/21 SPARTA portfolio review ORE Catapult (2022).

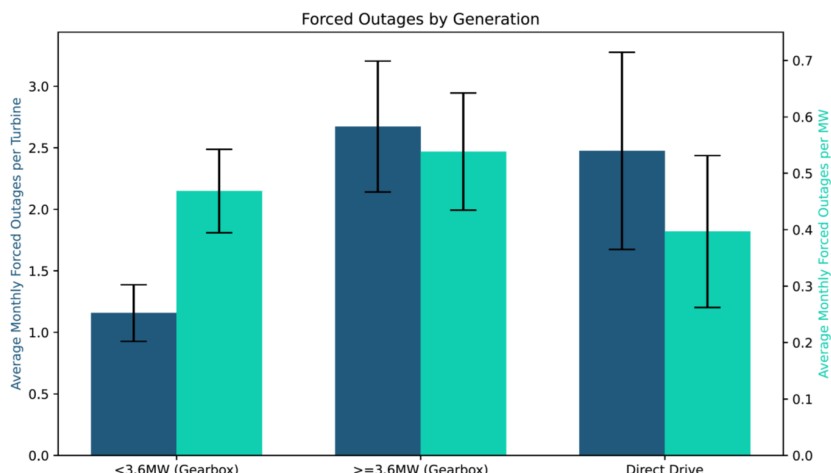

**Figure 9.** Comparison of different wind turbine concepts from the 20/21 SPARTA portfolio review ORE Catapult (2022).

works and failures per turbine per year in the case of corrective works. The study also presents mean downtime per mainte-
nance action. Likewise to those of Jenkins, Koukara & SPARTA, the dataset has some limitations. Firstly, the dataset is only

representative of one turbine model at one location, and so is much less representative of a generic wind turbine as the other
reliability figures. Second, the turbine model which is represented is a high-speed geared machine, so the figures are not di-
rectly applicable to the research question of this study. Third, Anderson highlights in his thesis that there are uncertainties in





the failure rate figures due to (i) the possibility for different failure definitions and (ii) the possibility for misinterpretation of work orders and alarm codes from the researcher.

### 2.0.5 Summary

This section presents a review of literature surrounding drive-train reliability for wind turbines since the publication by Carroll et al. (2016) and not included by Turnbull et al. (2022). These are summarised in Table 1. With the aim to complete a cost modelling comparison between direct-drive and medium-speed machines, the key components are (ostensibly) the generator, gearbox and main bearing. The level of detail for future-generation turbines to be taken away for each of these components is summarised below.

1. Gearbox. Jenkins (2022) provides estimates for replacement rates of next generation gearboxes. These estimates present a significant increase in reliability compared to the work of Carroll et al. (2016). Referring to Figure 9, the annual major replacement rate for gearboxes, as estimated by SPARTA, is approximately 0.015. This is within the uncertainty range presented by Jenkins (2022), and reinforces the trend of improving reliability for wind turbine gearboxes. Improved repair rate estimates for gearboxes might be assumed from Koukoura (2019). "Improved" is used here to simply say that the failure data is less out of date than those used by Carroll et al. (2017). However, the population of turbines making up those figures is young. The failure rates presented therefore might be obfuscated by failures that occur later in the component lifetime, or by more frequent early failures.

2. Generator. Again, Jenkins (2022) provides estimates for replacement rates of next generation generators, for both Direct Drive (DD) and Medium Speed (MS) geared turbines. Similar to the gearbox, there is a reliability improvement over the first generation of wind turbines represented by Figure 3. SPARTA's figure of approximately 0.005 is within the uncertainty limits for medium-speed machines, but outwith that of direct-drive machines. Mechanical repair rates for geared turbines can be updated by Koukoura (2019)'s figures for > 5 MW machines. Similar to above, the failure rates represent a young population of turbines.

3. Main bearing. Neither Carroll et al. (2016) nor Jenkins et al. (2022b) present failure figures for main bearings. As summarised by Hart et al. (2020), main bearings are often neglected in reliability analyses such as those presented by Carroll et al. (2016), Hahn et al. (2007), Spinato et al. (2009) and Wilkinson et al. (2011) either by (i) lumping the main bearing in with the gearbox of (ii) not including it at all (Koukoura (2019)). Koukoura provides figures for the main bearing of both direct drive and geared configurations which might be used for cost modelling purposes.

Jenkins et al. (2022a) provides estimates for drive train component replacements based on a structured method for elicitation which incorporates uncertainty quantification. Since replacements are a driver of costs, that study provides a solid base upon which to build the analysis. Koukoura (2019) data is useful for building on that base to provide updated reliability estimates for the drive train for offshore turbines which are less out-of-date than those used previously by Carroll et al. (2016). However, those figures will likely still not represent accurately the failure rates of a 15 MW wind turbine. Since they are only mechan-





185 ical failures, they will also need to be fleshed out by minor failure rates from Anderson (2023)- the dataset which is least representative of a future 15 MW machine, but nevertheless is still a new source of failure data.

**Table 1.** Summary of the data-tables relevant to a reliability analysis.

| O&M Data Type | Information Derived | Disadvantages |
|---|---|---|
| Jenkins et al. (2022b) | – Major replacement rates of next generation generators, gearboxes and rotors from expert elicitation<br>– Median, 5th and 95th percentile estimates<br>– Global and equal weighting methods. | – No main bearing replacements<br>– No Major repair/minor repair rates |
| Koukoura (2019) | – Mechanical repair/replacement rates for drivetrain assemblies;<br>– Comparison between geared and direct drive turbine failure rates;<br>– Comparison between sub-5MW and over-5MW failure rates | – No electrical failure rates in the generator;<br>– No repair times/number of repairs;<br>– Young population of direct drive turbines. |
| SPARTA ORE Catapult (2022) | – Major replacement rates of the BoP, blade, electrical, gearbox, generator and main bearing components;<br>– Forced outage rates;<br>– Comparison of forced outage rates between direct drive & geared turbines | – No repair rates;<br>– No repair times/ number of technicians;<br>– Young population of direct-drive turbines. |
| Anderson (2023) | – Repair and replacement rates for high-speed geared turbine assemblies;<br>– Major Replacement/Major repair/minor repair categorisation | – Older turbines (not medium speed or direct-drive)<br>– Only data from one wind farm. |





# 3 Methodology

## 3.1 O&M Model

This study relies on the StrathOW-O&M model for the O&M modelling. The model was developed at the University of Strath-
clyde by Dinwoodie and McMillan (2014). It was subsequently validated against three other cost modelling tools Dinwoodie
et al. (2015). Since then, it has been further developed and frequently utilised by other Strathclyde researchers Dalgic et al.
(2015a); Carroll et al. (2017); Flannigan et al. (2022). See Dinwoodie (2014) for a detailed description of the model.

StrathOW-O&M models a series of work shifts simulated in the time domain. Three simulated time-series feed into the
central simulation: one describing significant wave height and wind speed; one the ideal power production for the farm; and
another the probability of a subsystem failure in each time-step. These are all derived from user inputs describing the met-ocean
climate, power curve and turbine reliability estimates respectively. The weather conditions are generated using the historical
climate data set provided by the user. A Non-Homogeneous Poisson Process (NHPP) is used to model reliability through time.
For each time step, the conditional reliability of a subsystem is compared to a randomly generated number to determine if that
subsystem has failed. When a failure occurs, repairs are carried out dependant on availability of required resources (in terms of
vessels, staff and materials) and climate restraints on vessel operational usage (significant wave height and wind speed limits).
Again, the resource and operational limits are user-defined.

Once the shift is simulated, the model records the condition of the wind farm in terms of turbines available and resources
utilised. The process is repeated for the specified lifetime of the farm, and the lifetime power production and availability are
calculated and stored. This is repeated until there is convergence of availability estimates on cross-simulation values. Cross-
simulation calculations are passed to model outputs for post-processing. Outputs consist of a list of Key Performance Indicators
(availability, power production and number of failures), cost estimates (revenue, lost production costs, vessel and staff costs,
costs of spare parts) and vessel specific information (Crew Transfer Vessel (CTV) utilisation, number of Jack-Up Vessel (JUV)
charters).

## 3.2 Reliability Model Inputs

### 3.2.1 Failure Rates

Failure rates for every component bar the drive train are assumed to be equivalent for geared and direct-drive turbine concepts.
These are taken from Carroll et al. (2016), excluding the rotor major replacements rate, which is taken from Jenkins et al.
(2022b). Assumed failure rates for the gearbox, generator and main bearing are summarised in Table 2.

### 3.2.2 Repair Times and Required Number of Technicians

For the baseline comparison, repair times and number of technicians are based on the work by Carroll et al. (2016). Main
bearing replacements/repairs are assumed to take the same time as gearbox replacements/repairs and use the same number of
technicians.



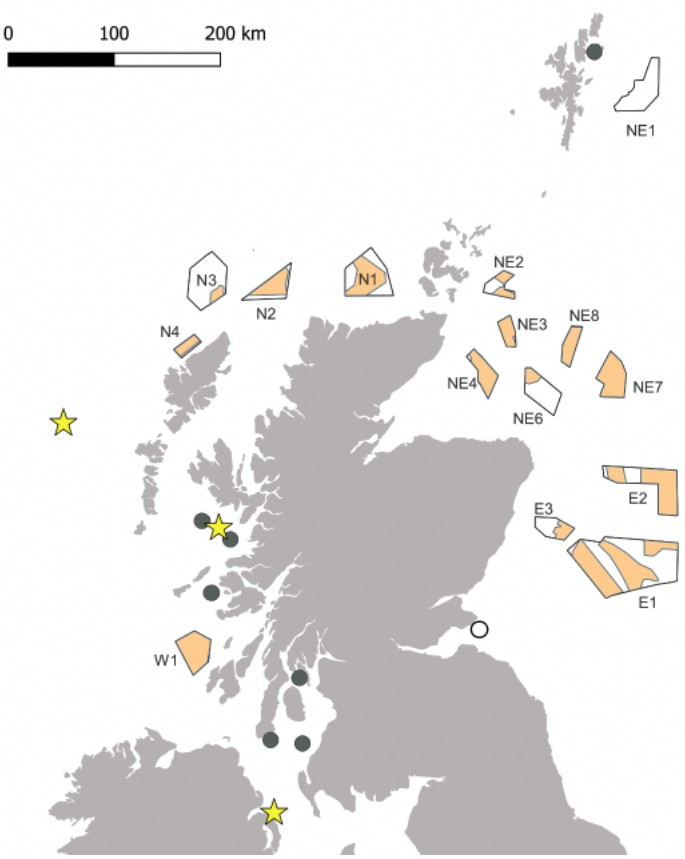

**Figure 10.** Map of Scotwind leasing sites Crown Estate Scotland (2023a). The three chosen sites of Bowdun, Caledonia and Machair are identified by the codes E3, NE4 and W1 respectively.

### 3.2.3 Repair Costs

Replacement costs are based on BVG's guide to an offshore wind farm (Catapult (2019)). Since those figures are representative
220  of 10 MW turbines, they are scaled to 15 MW turbines by power rating based on a linear cost assumption (i.e. all components are multiplied by 1.5). Repair costs are based on Carroll (2016) and calculated in a similar way to Flannigan et al. (2022). The cost ratios for minor and major repairs to major replacement costs are calculated. Those ratios are multiplied by the cost of a new component to estimate the repair cost.

### 3.3 Wind Farm, Power Curve and Weather Specification

225  Similarly to Jenkins et al. (2022a), a 1.5 GW wind farm with of 100 x 15 MW wind turbines is used. The farms are assumed to have a 25 year operational lifetime. Three of the locations designated as suitable for fixed-bottom sites in the Scotwind offshore wind leasing process were used as case studies (Crown Estate Scotland, 2023a). Namely, the Bowdun, Caledonia and



**Table 2.** Summary of baseline failure rate inputs.

|  |  | Medium-Speed |  | Direct Drive |  |
| --- | --- | --- | --- | --- | --- |
| Component | Failure Mode | Failure Rate | Source | Failure Rate | Source |
| Gearbox | Replacements | 0.022 | Jenkins | N/A | N/A |
|  | Planet Wheel Repair | 0.05 | Koukoura | N/A | N/A |
|  | Other Gearbox Minor Repair | 0.32 | Anderson | N/A | N/A |
| Generator | Replacements | 0.01 | Jenkins | 0.021 | Jenkins |
|  | Generator Bearing Replacement | 0 | Koukoura | 0 | Koukoura |
|  | Generator Bearing Repair | 0.196 | Koukoura | 0 | Koukoura |
|  | Other Generator Minor Repair | 0.22 | Anderson | 0.22 | Anderson |
| Main Bearing | Replacements | 0.009 | Koukoura | 0.015 | Koukoura |
|  | Main Bearing Repair | 0.006 | Koukoura | 0.062 | Koukoura |

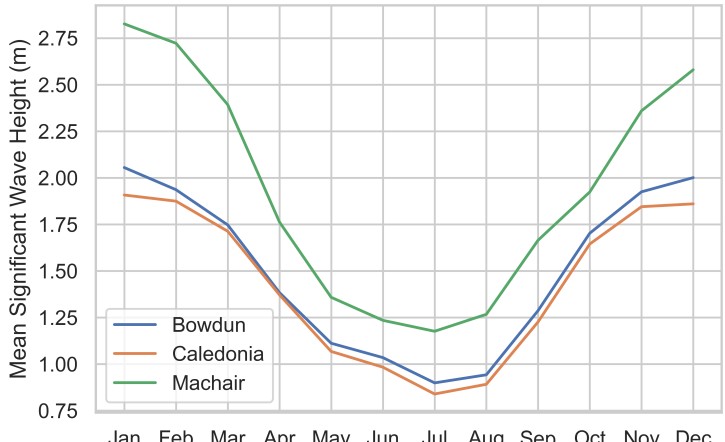

**Figure 11.** The mean significant wave height for the three selected ScotWind sites, revealing that Machair has on average a higher significant wave height than Bowdun and Caledonia sites.

Machair sites depicted in Figure 10. ERA5 reanalysis data for each site was downloaded from online ESOX tool developed by Lautec Eso (2023). This consists of twenty-five years of wind speed at a height of 100m and significant wave height values at an hourly frequency. For all case studies the power curve used is based on the IEA 15 MW reference turbine from Gaertner et al. (2020) and an electricity price of £40.7/MWh was assumed based on the work by Wiser et al. (2021).

An important site characteristic for the wind farms in this analysis is the accessibility. Accessibility is the measure of the amount of time that the climate at the site is below the accessibility limits, that are set by the marine coordinator, to allow the transfer of technicians to turbines for maintenance to occur. The two widely used limits for vessels is the wind speed and wave



height. Figure 11 shows the average significant wave height for the three chosen sites. For reference, the CTV access limit is 1.5 m and for the SOV is 2.5 m. Machair site has the highest average significant wave height throughout the months of the year with Caledonia and Bowdun at a similar lower average. The higher significant wave height values will potentially reduce accessibility for Machair.

### 3.4 Vessel Inputs

The vessel inputs are shown in Table 3. An combined SOV(Service Operation Vessel)-CTV-based strategy is assumed at the site. This is effectively modelled as a mothership concept as described by Dalgic et al. (2015b), where the 'mothership' vessel is assumed to have an access mechanism for technician-to-turbine transfers. No mooring capabilities are assumed to be in use. For an overview of the input parameters for the vessels, see Table 3. Note that, for some of the costs, we have rounded the figures when converting from pounds to euros.

**Table 3.** Summary of vessel inputs. Wave height limits are based on work by Dalgic et al. (2015a); McMorland et al. (2022) and charter rates come from Catapult (2019, 2020).

| Vessel Type | CTV | SOV | JUV |
|---|---|---|---|
| Number of Vessels | 4 | 1 | 1 |
| Charter Day Rate (€) | 3,000 | 30,000 | 420,500 |
| Mobilisation Time (Days) | N/A | N/A | 60 |
| Mobilisation Cost (€) | N/A | N/A | 2,100,000 |
| Charter Type | Continuous | Continuous | Fix-on-Fail |
| Charter Length (Days) | Lifetime | Lifetime | 60 |
| Wave Height Limit (m) | 1.5 | 2.5 | 2.8 |

## 4 Results and Discussion

The following section provides the simulation results for the baseline scenario that compares the three wind farm sites for the two drive train configurations. After simulating the baseline scenario, multiple sensitivity analyses are completed to determine the susceptibility of the cost outputs to several inputs.

### 4.1 Baseline Comparison Between Direct Drive & Medium-Speed

The results of the baseline comparison are shown in Figures 12 and 13. Figure 12 shows the baseline availability results. For all scenarios, the direct drive has higher availability than the medium-speed concepts. The difference in the absolute percentages of the two configurations is 0.49 %, 0.54 % and 1.73 % for the Bowdun, Caledonia and Machair sites respectively. The delta is notably greater for Machair, the site characterised by lower accessibility than the other two, as seen in Figure 11.




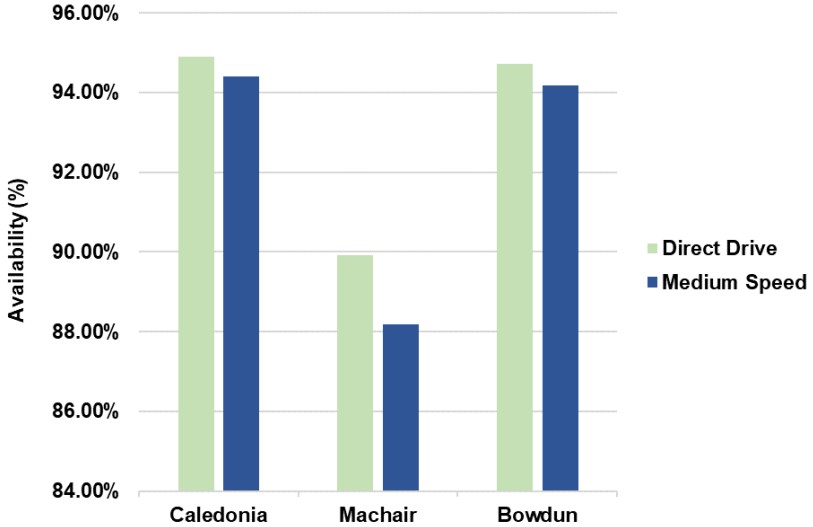

**Figure 12.** Comparison of availability for the three chosen wind farm sites between the two drive train configurations.

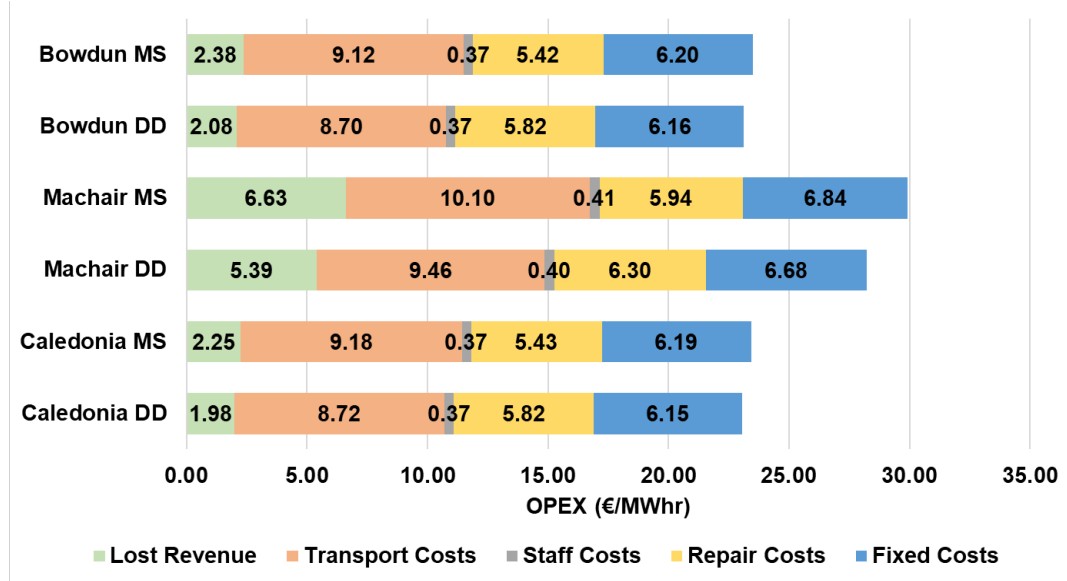

**Figure 13.** Breakdown of OPEX costs for the three wind farm sites for the 15 MW medium speed (MS) turbine scenario and the 15 MW direct drive (DD) turbine scenario.

Figure 13 shows the O&M cost results, measured in €/MWh. For all scenarios, the direct drive configuration has lower
255 O&M costs than the medium-speed. Again, the difference is greater for the less-accessible Machair site than the Bowdun and




Caledonia sites. The relative percentage difference between the two configurations is around 1.59 %, 1.58 % and 5.78 % for the Bowdun, Caledonia and Machair sites respectively.

Important differences in the cost breakdown between the two concepts are as follows:

1. **Lost Revenue costs**. Opportunity costs are higher for medium-speed turbines than direct-drive turbines. The medium-speed machines are still expected to have (i) higher overall failure rates and (ii) more major replacements than direct drive turbines. The relative difference between the two configurations, in terms of the lost production cost, is higher at the less-accessible Machair site (at around 20 %) compared to the Caledonia and Bowdun sites (at around 13 % each).

2. **Transport costs**. Vessel costs are higher for medium-speed turbines than direct drives. Since there are more major replacements for medium-speed machines, the increased utilisation of JUVs results in higher transport costs.

3. **Repair Costs**. Repair costs are higher for direct-drive turbines compared to medium-speed machines. This is attributable to higher assumed generator costs. According to BVG's guide to an offshore wind farm Catapult (2019), a direct-drive generator costs twice as much as that of a medium-speed. The cost of replacements and repairs are therefore assumed to be twice as much for direct-drive generators. This extra assumed cost is enough to exceed the repair costs afforded to the gearbox. This is enough of an expenditure that it outweighs the extra cost of gearbox repairs (at least under the current assumptions).

## 4.2 Sensitivity of Assumptions

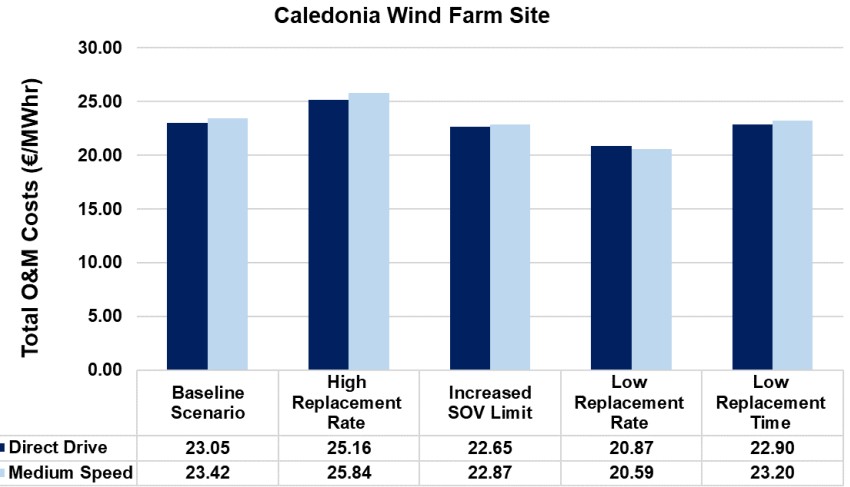

**Figure 14.** Sensitivity analysis results for Caledonia wind farm site. The total O&M costs of the baseline scenario are compared to the total O&M costs for each analysis.

Due to the nature of the inputs used for the model and the various sources utilised to synthesise the baseline scenario, it is important to investigate how sensitive these assumptions are. There are three inputs which were identified for sensitivity



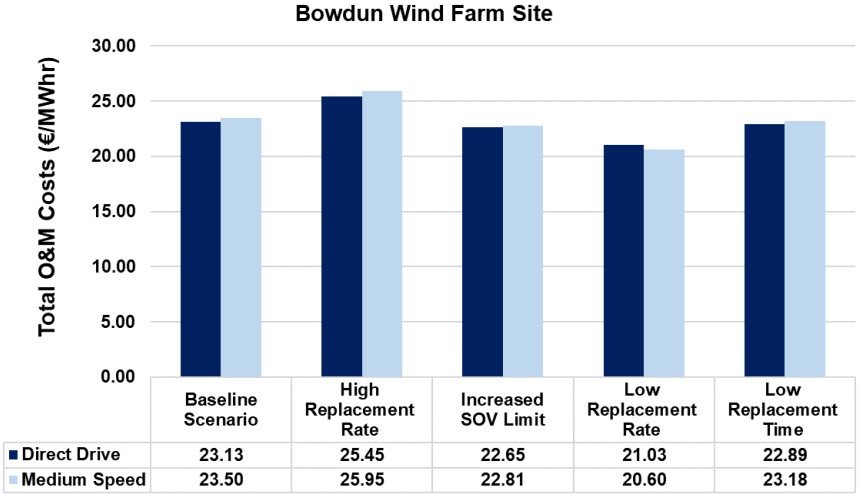

**Figure 15.** Sensitivity analysis results for Bowdun wind farm site. The total O&M costs of the baseline scenario are compared to the total O&M costs for each analysis.

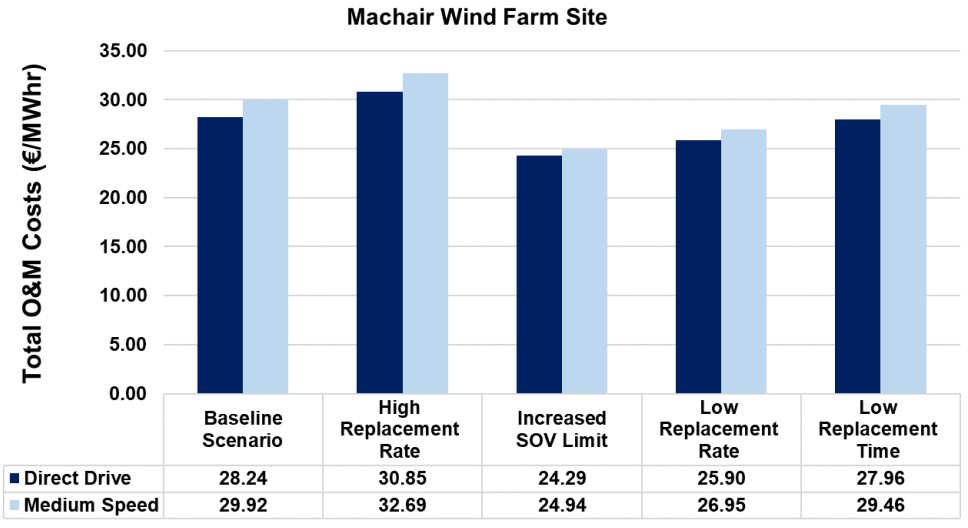

**Figure 16.** Sensitivity analysis results for Machair wind farm site. The total O&M costs of the baseline scenario are compared to the total O&M costs for each analysis.

analysis: replacement rate, SOV accessibility limits and the replacement times. For the replacement rates, two analyses were completed by using the 5th and 95th percentile estimates from Jenkins (2022), as a low replacement rate scenario and high replacement rate scenario. These were input into the model for the generator, gearbox and rotor components.




Secondly, the accessibility for the SOV was increased. The Machair site was identified as the least accessible site which, after analysing the baseline scenario, had a clear impact on the overall costs of the wind farm. The SOV wave height limit input was adjusted from 2.5 m to 3.5 m to increase accessibility for maintenance trips.

Additionally, the repair times for the baseline scenario are based off Carroll et al. (2016), but a study by Catapult (2020) uses the assumption that major replacements take under 47 hours to complete. Using this assumption, the repair times which exceeded 47 hours in the baseline scenario are set to 47 hours to simulate the lower replacement times. Finally, the repair costs were altered post processing. The repair costs were based on 10 MW turbine costs and scaled linearly by 1.5 times for the 15 MW turbines. In the sensitivity analyses the minor repair costs were scaled down to the original values, as it is not known that minor repairs would continue to increase in cost linearly with the turbine size. However, the results showed that altering the minor repair costs in this manner, resulted in less than a 1 % change in total O&M costs for all sites and the two different configurations. Therefore, it has not been included in the following figures as the change in repair costs was deemed not significant compared to the other factors being studied.

These four sensitivity analyses results are plotted in Figures 14, 15, 16 against the baseline scenario. The figures graph the total O&M costs for each of the wind farm sites. To highlight the change in costs from the baseline scenario, Table 4 shows the % change for the three wind farm sites.

One of the main takeaways from Figure 14, is that the direct drive is the drive train configuration with lower costs for all cases except for the low replacement rate at the Caledonia wind farm. The same trend is seen for the Bowdun wind farm in Figure 15, where direct drive has higher costs than medium speed for the low replacement rate case. Table 4 indicates that for the higher replacement rate, the cost difference between the configurations becomes wider, with medium speed costs increasing more than the direct drive costs. Aside from the high replacement rate case, all other scenarios result in the reduction in the cost gap between the medium speed and direct drive case, which, in two scenarios resulted in the direct drive having higher cost than the medium speed. For the Machair site, the high replacement rate resulted in the medium speed still having higher costs than the direct drive as seen in Figure 16. However, in Table 4, it is clear that the % increase in costs compared to the baseline was 9.26 % for medium speed, only a slightly bigger increase than the 9.23 % for the direct drive.

**Table 4.** The % change in total operational costs from the baseline scenario for all three wind farm sites. Note that 'High Replacement Rate' values are a percentage increase in the operational costs whereas 'Low Replacement Rate', 'Increased SOV Limit' and 'Low Replacement Time' values are a % decrease in operational costs. 'DD' represents the Direct Drive turbine simulation and 'MS' represents the Medium Speed turbine simulation.

|  | Caledonia | | Machair | | Bowdun | |
|---|---|---|---|---|---|---|
|  | DD | MS | DD | MS | DD | MS |
| High Replacement Rate | 9.16 | 10.31 | 9.23 | 9.26 | 10.00 | 10.43 |
| Low Replacement Rate | 9.43 | 12.09 | 8.27 | 9.93 | 9.07 | 12.31 |
| Increased SOV Limit | 1.72 | 2.38 | 14.00 | 16.64 | 2.08 | 2.92 |
| Low Replacement Time | 0.63 | 0.95 | 0.98 | 1.53 | 1.03 | 1.35 |





### 4.2.1 High Replacement Rate

In this analysis, the failure rates were increased to the 95th percentile estimate of the baseline scenario failure rates for the major components of the drive train for both configurations. This resulted in higher operational cost for all sites due to the increase on repairs required. Based on Table 4, the % increase in costs was relatively similar in all scenarios, between 9-10.5%

increase in the operational costs compared to the baseline scenario. The high replacement rates resulted in a larger % increase in total operational costs for medium speed configuration for all three sites. For the Machair site, there was a similar % increase for the total operational costs for both configurations, therefore there was little change to the cost gap between turbines. This could be related to the accessibility issues at the site.

### 4.2.2 Low Replacement Rate

Lowering the failure rate for the major components of each turbine to the 5th percentile estimate from the baseline scenario failure rates resulted in a decrease in operational costs for all sites. The reduction in failures occurring leads to a reduction in downtime, which contributes to lower lost revenue, as well as lower transport costs, repair costs and staff costs. In all cases, there was a significant % reduction in costs compared to the baseline case both for medium speed and direct drive, as seen in Table 4. However, the % reduction in costs was larger for the medium speed turbines than the direct drive turbines, resulting

in a smaller difference in cost between the configurations compared to the difference seen in the baseline scenario. The reason behind the larger % reduction for medium speed is that the failure rates for medium speed that were lowered also included the gearbox component which is a large source of failure for the turbine. Reducing the failure for this component, significantly reduces the overall cost, whereas the drive train for the direct drive does not have a gearbox and so does not benefit.

### 4.2.3 Increased SOV Limit

From Table 4, all scenarios had a reduction in the total operational cost compared to the baseline case when the SOV limit was increased from 2.5 m to 3.5 m. Increasing the accessibility of a site will reduce the lost revenue generated due to longer down times, explaining the reduction in cost seen for this analysis. In particular, the Machair site saw a large % reduction in operational costs which stems from the low accessibility the site had in the baseline scenario, so altering the SOV limit would benefit this site more than the other two site selections. In terms of configurations, the medium speed configuration saw

a greater reduction in the total operations costs for all three sites when compared to the direct drive turbine case. Increasing the SOV wave height limit benefits medium speed turbines more as these turbines have higher failure rates and an increased number of major replacements required, due to the additional gearbox component. Therefore, increasing accessibility will reduce a greater amount of downtime for medium speed, that leads to the greater % difference in cost between the baseline scenario and the increased SOV limit scenario. In this analysis, the increased limit for SOV transfers also results in a smaller

total difference in operational costs between the configurations which narrows the gap between the two drive train types.





### 4.2.4 Low Replacement Time

Setting the maximum replacement time to 47 hours meant a number of components had a reduced time required for replacement. As a result, the operational cost for all sites was lowered when compared to the baseline scenario. The reduction in total operational costs was not as significant as the other sensitivity analyses, indicating that replacement time is not as significant a
factor in the operational costs as the failure rates or the accessibility limits of the wind farm. There was, however, a larger % reduction in costs for the medium speed turbines than the direct drive turbines in this analysis. For all sites, the medium speed saw a larger reduction which stems again from the gearbox component. The gearbox replacement time in the baseline scenario is 231 hours, so lowering this time to 47 hours saw a larger reduction in the overall costs than the direct drive which does not avail of the gearbox component.

## 5 Discussion

### 5.1 Limitations of Analysis/Model

Operations and maintenance modelling comes with a level of uncertainty, as with all models they are never a perfect representation of reality. This analysis also holds this uncertainty in certain aspects. Firstly, the reliability figures taken from Koukoura (2019) are based on two different graphs presented in the work, there may be a level of uncertainty in using the two different
sources. It is also assumed that these failure rates can be applied to a 15 MW turbine which may not be the case. However, the assumption is that even with a 10 MW power curve the trend between configurations would not differ but the magnitude of the power produced and therefore the magnitude of the costs may change. As seen in Table 2, some of the generator failure rates are 0, this may be due to the fact the data is taken from a relatively young population of turbines which have not had failures, adding to the uncertainty of some of the results. The reliability data from Section 2 is taken from four different sources that
categorise failure data in different ways. Therefore, there is a possibility that the failure rates for the components for minor repair, major repair and major replacement may have some overlap. Furthermore, the reliability figures from Anderson (2023) for the generator and gearbox minor repair and major repair come from high speed geared machines and do not directly translate to direct drive and medium-speed turbines. Other model inputs are based on previous literature and industry knowledge, however, it is important to consider that some of the literature may be considered outdated and therefore the inputs may carry
a level of uncertainty. This study assumes that the the only failure rates that change between the two configurations are the gearbox, generator and the main bearing. However, the work done by Turnbull et al. (2022) suggest that this is not the case. The study showed that direct drive turbines had twice as many stops per year compared to the medium speed geared turbine. Further work would involve adjusting some of the direct drive components based on the work done by Turnbull et al. (2022) to see the impact this would have on the cost gap between the configurations.
The model aims to capture the most important aspects of O&M modelling but due to the complex nature of the industry itself, it is impossible to accurately capture the variability of all the different aspects. In this analysis, some of the factors that may not be true to reality are:





1. Electricity prices. The electricity prices of the UK are constantly changing and so the fixed price chosen for this study may be subject to change in reality.

2. Future failure rates. Future failure rates are still unknown, therefore as time progresses the inputs used in this analysis may prove to not match to the operational data of future wind farms.

3. The supply chain. Finally, the supply chain is becoming a growing concern for a lot of wind energy experts who have stated that the expanding industry may not have the resources and vessels to support the growth. Supply chain bottle necks are not captured in this O&M model although this may be an important consideration in reality.

**5.2 Key Takeaways**

In the majority of cases, this study corroborates Carroll (2016)'s finding that direct drive turbines outperform medium speed turbines in terms of availability and O&M costs. However, considering more recent failure rate estimates for offshore turbines leads to a narrower gap than that presented by Carroll (2016), where accessibility is good. The work done by Carroll (2016) found the difference in availability between configurations, for a site 50 km from shore, to be roughly 0.7 %, with direct drive

having the higher availability. The baseline case in this study finds the difference in availability to be 0.49 % and 0.54 % for Bowdun and Caledonia, which are also 50km from shore. Previously, in terms of cost, Carroll (2016) found the absolute % difference in O&M costs for the two configurations to be 29.79 %, with direct drive having lower cost. Whereas, with the updated reliability data used in this analysis, the absolute % difference between the configurations for the baseline comparison were 1.58 %, 1.59 % and 5.78 % for Bowdun, Caledonia and Machair respectively. Medium speed turbines incur larger O&M

costs if the site suffers from low accessibility, due to the increased amount of major replacements and higher overall failure rates. This was highlighted in this study through the Machair case study site. At a certain threshold of improved drive train reliability, there are scenarios where the medium speed turbines have lower operational costs than the direct drive turbines. These lower operational costs are dependent on the accessibility of the site. One way to tackle the higher costs incurred for medium speed turbines at low accessible sites, is by increasing the wave height limit for vessels to allow more transfers to

occur. While the results from this analysis paint the medium speed turbines in a more favorable light than previous studies, it is important to note that the direction of industry is to select sites further from shore which tend to have lower accessibility.

Similarly, if future generation turbines have higher failure rates than expected in equal measures across both drive train configurations, based on the analysis carried on in this study, the operational costs for the medium speed will be higher than the operational costs of a direct drive turbine and more importantly, that cost gap between the configurations will be larger.

One factor that did not seem to have as large an impact on the operational costs of a site is the replacement time for major components. Although, lower replacement time did lower costs and lower the cost gap between configurations, it had the least impact on costs when comparing to failure rates and accessibility limits. The apparent reduction in cost gap between the two configurations seems to corroborate with the opinion held in industry. Many developers are still investing in two stage medium speed turbines and direct drive turbines which implies an understanding that both turbines are viable options for offshore wind





farms. Whether this trend continues, remains to be seen as there is still a large uncertainty surrounding the operation of larger rated turbines for configurations.

## 6    Conclusion

The motivation for this study was to determine if the conclusions drawn for previous reliability studies for smaller rated turbines applied to the future generation of larger rated turbines (15 MW). Previous reliability studies found that direct drive
turbines provided lower operational costs and higher availability than medium speed geared turbines. Using new sources of failure rate data and synthesising these for minor repair, major repair and major replacement for the main components of the turbine, namely the generator, gearbox and main bearing, the operations and maintenance for a wind farm with 15 MW turbines was simulated. Simulations were ran using the Strath OW O&M modelling tool, which took the new failure rate data and updated costs for repairs, and produced total operational costs and availability for three different case study wind farm
sites. The simulations showed that the direct drive turbines for all three sites produced lower operational costs than the medium speed turbines as well as a higher availability. However, in comparison to previous studies, the cost gap between these two configurations was much smaller. Once the baseline scenario was complete, several sensitivity analyses were undertaken to determine the impact the inputs had on the overall costs. In a high replacement rate scenario, costs increase by roughly 10 %, whilst a low replacement rate scenario results in a 10 % decrease in costs, for all wind farm sites for both configurations.
The increase of replacement rates effected the cost of the medium speed turbine more than the direct drive, resulting in an increase in the cost gap between configurations. The decrease of the replacement rate resulted in the opposite, with two sites having a higher cost for the direct drive over the medium speed. The least accessible site was most sensitive to the adjustment of the SOV wave height limit. When the SOV wave height limit was increased, allowing more transfers, the site saw a 16 % decrease in costs for the medium speed and a 14 % decrease in costs for the direct drive, in comparison to the baseline case.
The replacement rate was also lowered for a sensitivity analysis but this had the least impact on the operational costs for the sites. Ultimately, the takeaway from the study is that, in the case that accessibility for the site is low, the direct drive turbines produce lower operational costs than the medium speed geared turbines for next generation turbines rated at 15 MW. This conclusion is mainly drawn from the fact that the medium speed turbines have higher failure rates and require more major replacements due to the gearbox component and so if accessibility is low, the downtime for that wind farm will be larger than
a direct drive turbine wind farm. If there is good accessibility for the site then the cost gap between configurations is much smaller. Furthermore, a medium speed turbine may have lower operational costs than a direct drive turbine if the failure rates are below a certain threshold, in the case of this study, the 5th percentile estimates from Jenkins (2022). These findings are important as developers begin to plan for the future, where wind farms will contain larger rated turbines and they need to determine which drive train is best suited to the chosen site. It is suggested, moving forward from this study, that focus is
drawn on obtaining more reliability data from larger operational turbines so that uncertainties in the modelling of operations and maintenance for offshore turbines can be reduced. To build upon this study, obtaining updated failure rates for components other than the generator, gearbox and main bearing would allow a more complete analysis to be carried out.



**Data Availability**

The datasets used for the climate modelling in this work comes from the ERA5 reanalysis data and downloaded using the
ESOX tool by Lautec Eso (2023) and information on how to access the data can be found at https://esox.lautec.com/.

*Author contributions.* Orla Donnelly: Formal Analysis, Writing. Fraser Anderson: Data curation, Writing - Original draft preparation. James
Carroll: Supervision.

*Competing interests.* The authors declare that they have no conflict of interest.

*Acknowledgements.* Acknowledgement to the continued support from the UKRI Engineering and Physical Sciences Research Council EP-
SRC (Project EP/S023801 and Project EP/T031549/1).



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
