# Peer review of "Operations and Maintenance Cost Comparison Between 15MW Direct-Drive and Medium-Speed Offshore Wind Turbines"

_Wind Energy Science, 2023_

## Author Comment (AC1)

**Response to Reviewer 1:**

Firstly, thank you to the reviewer for taking the time to review and provide comments for this manuscript we appreciate it. We will address the reviewer's comments which are in **black bold** font with our response in *italics*.

**The manuscript analyzed the operational costs between different configurations of 15 MW turbines by summarizing existing literatures, performing simulations, and conducting sensitive analyses on various scenarios.**

**Overall, the topic and analysis of the manuscript is of a great interest of the wind energy science/industry. The procedure and results of the research is well organized and presented. The conclusion of "a medium speed turbine may have lower operational costs than a direct drive turbine if the failure rates are below a certain threshold" will be a good guidance for wind farm engineers to plan the future and brings benefits and efficiency to the wind energy industry.**

*Thanks to the reviewer, we appreciate the positive feedback and agree that this work will hopefully be useful for future studies and provide some insight into the operations and maintenance of larger turbines in the future.*

**However, I have several comments that will improve the manuscript if addressed before publication:**

- **In line 50: it would be better to provide a 1-2 sentences brief introduction of each configuration.**

  *Thanks for the suggestion, in the revised manuscript we have added the following lines to provide more context about the two configurations, 'Carroll et al (2017) provide a comparison of O&M costs for offshore wind turbines with different drive-train configurations. The chosen configurations represented the most popular offshore wind turbines at time of writing and varied from geared to direct drive configurations with different types of generators and converters. A geared turbine uses a gearbox, typically 2-stage or 3-stage, in it's drive train. The component sits between the turbine shaft connected to the rotor and the generator. It's role is to convert high torque low speed of the turbine shaft to low torque high speed for the generator. The downsides of this component are higher failure rates, that can be detrimental in offshore wind, as component repairs require heavy lift vessels incurring greater operational costs. The alternative to a gearbox is a direct drive configuration which removes the gearbox component entirely. The downside to this configuration is the heavy and expensive generators required in absence of a gearbox to convert high torque. To elaborate, the configurations studied in Carroll et al (2017) are:'*

- **I found that the Figure 1 is not very informative since the two figures are very similar and the figure description is not providing any extra information other than describing the figures. It would help readers to have a better understanding if the figure can focus on the parts (i.e., the gearbox and the direct drive) with differences.**

  *Figure 1 has been altered slightly and the size of the generator has been increased for the direct drive configuration. We agree that it previously lacked context. The updated figure now paired with the lines added after line 50 will hopefully address the reviewer's suggestion.*

- **The 5.2 Key Takeaways session can be reworked to bullet points, which helps readers to get a clearer view of the numbers and comparisons.**

*Taking this on board we have summarised this section into key bullet points, we have not made major changes to the takeaways themselves as it feels important to include as much information as possible for the reader for this section in particular but agree that the bullet points make it easier to separate the individual points. The changes are as follows,*

*'In the majority of cases, this study corroborates Carroll et al.(2017)'s finding that direct drive turbines outperform medium speed turbines in terms of availability and O\&M costs. However, considering more recent failure rate estimates for offshore turbines leads to a narrower gap than that presented by Carroll et al.(2017)'s, where accessibility is good.*

*To summarise, the main points of interest to take away from this study are:*

- *The work done by Carroll et al.(2017)'s found the difference in availability between configurations, for a site 50 km from shore, to be roughly 0.7 %, with direct drive having the higher availability. The baseline case in this study finds the difference in availability to be 0.49 % and 0.54 % for Bowdun and Caledonia, which are also 50 km from shore.*

- *Previously, in terms of cost, Carroll et al.(2017)'s found the absolute % difference in O&M costs for the two configurations to be 29.79 %, with direct drive having lower cost. Whereas, with the updated reliability data used in this analysis, the absolute % difference between the configurations for the baseline comparison were 1.58 %, 1.59 % and 5.78 % for Bowdun, Caledonia and Machair respectively.*

- *Medium speed turbines incur larger O&M costs if the site suffers from low accessibility, due to the increased amount of major replacements and higher overall failure rates. This was highlighted in this study through the Machair case study site.*

- *At a certain threshold of improved drive train reliability, there are scenarios where the medium speed turbines have lower operational costs than the direct drive turbines. These lower operational costs are dependent on the accessibility of the site. One way to tackle the higher costs incurred for medium speed turbines at low accessible sites, is by increasing the wave height limit for vessels to allow more transfers to occur. While the results from this analysis paint the medium speed turbines in a more favourable light than previous studies, it is important to note that the direction of industry is to select sites further from shore which tend to have lower accessibility.*

- *Similarly, if future generation turbines have higher failure rates than expected in equal measures across both drive train configurations, based on the analysis carried on in this study, the operational costs for the medium speed will be higher than the operational costs of a direct drive turbine and more importantly, that cost gap between the configurations will be larger.*

- *One factor that did not seem to have as large an impact on the operational costs of a site is the replacement time for major components. Although, lower replacement time did lower costs and lower the cost gap between configurations, it had the least impact on costs when comparing to failure rates and accessibility limits.*

*Overall, the apparent reduction in cost gap between the two configurations seems to corroborate with the opinion held in industry. Many developers are still investing in two stage medium speed turbines and direct drive turbines which implies an understanding that both turbines are viable options for offshore wind farms. Whether this trend continues, remains to be seen as there is still a large uncertainty surrounding the operation of larger rated turbines for configurations.'*

- **The current conclusion session is a bit lengthy and unstructured. It would help the reader to get a clearer view of the important conclusions of this study. For example, the conclusion session can be easily rewritten into different paragraphs such as motivation of the study, conclusion of the simulation work, performed sensitive analyses and corresponding observations under different scenarios, and the importance of the findings and outlook, etc.**

*Thank you for the suggestion, the conclusion is long and I think due to the length of analysis and results it does become quite lengthy but we have reduced this as much as possible and we feel the conclusion is more concise than previously. We agree the structuring could be better and so, without trying to remove any of the key points of the conclusion, it is worded as follows:*

*'The primary objective of this study was to assess whether conclusions derived from prior reliability studies on smaller turbines could be extrapolated to the emerging generation of larger turbines, specifically those rated at 15 MW. Earlier research had established that direct drive turbines exhibited superior cost-effectiveness and higher availability compared to medium speed geared turbines. To investigate the applicability of these conclusions to larger turbines, the study synthesized new failure rate data for the generator, gearbox, and main bearing, simulating the operations and maintenance of a wind farm featuring 15 MW turbines. Utilizing the Strath OW O&M modelling tool, simulations were conducted for three distinct case study wind farm sites, incorporating updated repair costs.*

*The simulations consistently demonstrated that direct drive turbines, across all three sites, yielded lower operational costs and higher availability in comparison to medium speed turbines. Notably, the cost gap between the two configurations was found to be narrower than in previous studies.*

*Following the establishment of a baseline scenario, sensitivity analyses were conducted to evaluate the influence of various inputs on overall costs. In scenarios with high replacement rates, costs increased by approximately 10%, whereas low replacement rates resulted in a 10% cost decrease for both turbine configurations across all sites. The impact of replacement rate variations was more pronounced on medium speed turbines, widening the cost gap between the two configurations. Accessibility proved critical, with low-accessibility sites favouring direct drive turbines due to their lower failure rates and reduced need for major replacements, minimizing downtime.*

*The study's key takeaway was that, in situations where site accessibility is limited, direct drive turbines prove more economically viable for next-generation 15 MW turbines. However, with good accessibility, the cost gap between direct drive and medium speed configurations is reduced. Moreover, medium speed turbines might have lower operational costs than direct drive turbines if failure rates fall below a specific threshold, as indicated by the 5th percentile estimates from Jenkins et al. These findings hold significance for developers planning future wind farms with larger turbines, aiding them in selecting the optimal drive train for specific sites.*

*The study recommends a continued focus on obtaining reliability data from larger operational turbines to enhance the accuracy of operations and maintenance modelling for offshore turbines. Additionally, obtaining updated failure rates for components beyond the generator, gearbox, and main bearing is suggested for a more comprehensive analysis and informed decision-making in turbine selection.'*

- **Small formatting comments:**

  o **The unit "MW" should not be italic in item 2 above line 115.**

    *This has been changed, thanks.*

  o **BVG is not mentioned with full name in the manuscript when first occurring above line 220.**

    *We acknowledge that standard practice is to use the full name of any abbreviations before using the abbreviation but in this case the name of the company is BVG Associates and does not go by a full name or use that name in any publications/online. In this case we haven't changed this but instead have changed the name to 'BVG Associates' and hope our explanation as to why is sufficient.*

**I believe addressing these comments will improve the readability of the manuscript and enhance its contribution to the field. Overall, the research work has done a great contribution to the wind turbine industry.**

---

## Author Comment (AC2)

**WES-2023-156**

**Response to Reviewer 2:**

Firstly, thank you to the reviewer for taking the time to review and provide comments for this manuscript we appreciate it. We will address the reviewer's comments which are in **black bold** font with our response in *italics*.

**The manuscript entitled "Operations and Maintenance Cost Comparison Between 15MW Direct-Drive and Medium-Speed Offshore Wind Turbines" deals with a very interesting topic, which is very important for the near future development of wind turbine technologies.**

**In a nutshell, the authors analyze the O&M costs of 15 MW direct-drive and medium-speed offshore wind turbines. An O&M model named StrathOW (which I guess has been developed at the University of Strathclyde) is employed to simulate scenarios and the input of the model is given by reliability data for the various sub-components which are taken from the literature. A sensitivity analysis is conducted and extreme cases are contemplated as well.**

*Thank you, Davide, for taking the time to review this submission to Wind Energy Science. We appreciate the positive feedback you have given and will address any suggestions to improve the manuscript below.*

**The quality of the presentation is very good and my comments to the paper are minor.**

**1) I would appreciate some more details about the application of the O&M model. It would be interesting for me to understand more in deep how much representative the results are.**

*You are correct that the model was developed at the University of Strathclyde. The application of the model is seen with our industrial partners who utilise this model for operations and maintenance modelling at present. As with all operations and maintenance models, they are never 100% representative, as the nature of O&M is that results are variable dependent on their inputs and a model will always carry some uncertainties around the results. However, we feel that this model captures important aspects and importantly highlights the difference between the two configurations. The values of cost are not necessarily the most important but rather the cost difference between the configurations. For a more in-depth understanding of the model and how it was benchmarked, Dinwoodie et al. have a paper, 'Reference Cases for Verification of Operation and Maintenance Simulation Models for Offshore Wind Farms' that provides more information (*https://doi.org/10.1260/0309-524X.39.1.1*)*

**2) Summarizing drastically, the result of the paper is that the former or the latter technology is slight more advantageous depending on a series of factors, which the authors discuss. I think that the wind turbine practitioners community might appreciate some more concrete guidelines, or at least criteria. Thus, I recommend to make an effort on this point.**

*We agree that it would be extremely beneficial if there were a set of concrete guidelines for developers but as alluded to before, operations and maintenance is a multi-faceted and due to multiple variables, often conclusions have to be more nuanced than concrete criteria. However, we acknowledge the reviewers point to make this clearer in our conclusion and so we have rewritten a section to highlight what the key takeaway for developers is, 'The study's key takeaway was that, in situations where site accessibility is limited, direct drive turbines prove*

*more economically viable for next-generation 15 MW turbines. However, with good accessibility, the cost gap between direct drive and medium speed configurations is reduced. Moreover, medium speed turbines might have lower operational costs than direct drive turbines if failure rates fall below a specific threshold, as indicated by the 5th percentile estimates from Jenkins et al. These findings hold significance for developers planning future wind farms with larger turbines, aiding them in selecting the optimal drive train for specific sites.'*